# An Increase in Solar Radiation in the Late Growth Period of Maize Alleviates the Adverse Effects of Climate Warming on the Growth and Development of Maize

Zhongbo Wei [1,†], Dahong Bian [1,†], Xiong Du [1], Zhen Gao [1], Chunqiang Li [2], Guangzhou Liu [1], Qifan Yang [1], Aonan Jiang [1] and Yanhong Cui [1,*]

1   State Key Laboratory of North China Crop Improvement and Regulation, Key Laboratory of Crop Growth Regulation of Hebei Province, College of Agronomy, Hebei Agricultural University, Baoding 071001, China; dr.weizhongbo@gmail.com (Z.W.)
2   Hebei Provincial Institute of Meteorological Sciences, Hebei Provincial Key Lab of Meteorological and Eco-Environment, Shijiazhuang 050021, China
*   Correspondence: cyh@hebau.edu.cn
†   These authors contributed equally to this work.

**Abstract:** Against the background of long-term climate change, quantifying the response of maize growth and development to climate change during critical growth stages will contribute to future decision-making in maize production. However, there have been few reports on the impact of climate change on maize dry matter accumulation and yield formation using long-term field trial data. In this study, field trial data from 13 agricultural meteorological stations in the Beijing–Tianjin–Hebei region from 1981 to 2017 were analyzed using partial correlation analysis and multiple regression models to investigate the effects of climate change on maize growth and yield. The results showed that the average temperature (Tavg) and accumulated effective precipitation (EP) during the maize growing season increased while the accumulated solar radiation (SRD) decreased from 1981 to 2017. During the seedling stage (GS1, VE-V8) and ear development stage (GS2, V8-R1), Tavg increased by 0.34 °C and 0.36 °C/decade, respectively, and EP increased by 1.83 mm and 3.35 mm/decade, respectively. The significant increase in Tavg during GS1 was the main reason for the inhibitory effect of climate change on maize growth, development, and biomass accumulation. However, the increase in SRD during the grain formation stage (GS3, R1–R3) and grain filling stage (GS4, R3–R6) was favorable for yield formation, increasing the grain number per ear (GN) and grain weight (GW) by 5.00% and 2.84%, respectively. SRD significantly increased after the silk stage, partially offsetting the adverse effects of temperature on maize yield formation, but the final result was a 0.18% and 0.94% reduction in maize plant dry weight (TDW) and grain yield (GY), respectively, due to the combined effects of the three climate factors. Spatially, climate change mainly had a negative impact on maize in the eastern and western parts of the central region of Beijing–Tianjin–Hebei, with a maximum GY reduction of up to 34.06%. The results of this study can provide a scientific basis for future decision-making in maize production against the background of climate change.

**Keywords:** maize; climate change; dry matter accumulation; various growth stages; growth and development

## 1. Introduction

The report from the Intergovernmental Panel on Climate Change (IPCC) points out that from 1983 to 2012, the average temperature was the highest in the past 1400 years, and climate change poses a serious threat to food security worldwide [1–3]. Several studies have indicated that due to global warming, the shortened crop phenology period has reduced crop yields by decreasing access to sunlight, water, and nutrients [4–7]. To adapt to climate change, previous researchers have explored the spatiotemporal changes in climate

resources during crop growth periods and used this information to optimize techniques and policies for crops to adapt to climate change [8,9]. Furthermore, they have adjusted the crop phenology period through breeding new varieties and adjusting planting times, aligning crop demands with the spatiotemporal distribution of climate resources to increase their production capacity [10–13]. In summary, these studies demonstrate that maximizing the use of climate resources is critical for maintaining stable and high crop yields.

Corn is the largest cereal crop in China, with an annual production that ranks second in the world [12]. Ensuring a stable corn yield in China is of significant importance for global food security. Studies have shown that climate change has contributed up to 40% of the reduction in China's corn yield, with reduced radiation and increased temperature during the growth period being the primary factors for the decline in corn productivity [14,15]. Wheeler et al. [16] argue that crop yields are more sensitive to high temperatures during the key growth and developmental stages of the crop. Li et al. [17] found that the impact of heat stress on the corn yield varies significantly depending on the growth stage. Vâtcă et al. [18] suggest that the evaluation of the impact of climate change on corn yield should be based on phenology and take into account the climate resource requirements of corn at each growth stage. These studies indicate that the degree of climate change impact on yield not only depends on the extent of climate factor change, but also on which growth stage of corn is most affected by climate change.

Using statistical models and historical data can help us to accelerate our understanding of crop growth and productivity in response to climate change [19,20]. Increasingly, studies worldwide are using mathematical and statistical methods to explore the impact of changes in heat resources on maize yield [21–23], and a small number of studies have compared the relative importance of different climate factors on maize yield [24,25]. However, these studies mainly focus on revealing the impact of seasonal climate change on maize yield and do not explicitly address the effect of climate change at different growth stages. Therefore, in recent years, some studies have explored the relationship between climate change at different growth stages and maize yield. Deng et al. [26] used regression models to assess the impact of climate change on maize yield in northeastern China based on trends in climate during the nutrition and reproductive stages of maize, finding that an increase in the minimum temperature during the nutrition stage contributed to an increase in maize yield. Zhu et al. [27] used monthly meteorological data on temperature, precipitation, and provincial maize yield data to construct a regression model, revealing that the effects of temperature and precipitation vary between different growth stages of maize. Chen et al. [28] used maize yield data and multiple climate factors to investigate the impact of climate change at different growth stages on maize yield in Sichuan, China, finding that climate change at different growth stages led to a significant change in maize production of approximately 30% in Sichuan province. However, most of these studies used fixed date divisions to classify growth stages and did not comprehensively consider the inter-annual variation in maize growth stages due to adjustments in sowing time or the use of new varieties, which could lead to incorrect assessments of trends in climate resource changes during growth stages. In addition, analyzing changes in maize yield and above-ground biomass accumulation under climate change can more comprehensively reflect maize's adaptation to climate change [29,30]. However, due to a lack of long-term targeted observations on crop trial data, few studies have analyzed the impact of climate change on maize biomass accumulation traits and yield traits. Therefore, the impact of climate change on maize growth and development at different growth stages is still largely unclear.

To clarify the changes in different climate factors during each growth stage of maize and elucidate the responses of maize growth and yield to the changes of different climate factors during different growth stages, long-term field trial data and meteorological observations were analyzed to investigate the relationship between climate change and maize growth and yield. This study aims to provide a theoretical basis for guiding maize production to cope with predictable climate change.

## 2. Materials and Methods

### 2.1. Study Area and Data

The study area covers Beijing, Tianjin, and Hebei Province, located between 36°05′–42°37′ N and 113°11′–119°45′ E. In this study, 13 representative agricultural meteorological experimental stations established and managed by the China Meteorological Administration (CMA) within this area were selected (Figure 1). The purpose of these agricultural meteorological experimental stations is to investigate local crop production and agricultural meteorological changes, and provide planting management advice to local farmers. They differ in geography and climate, and have good records of weather and crop data from 1981 to 2017.

The daily weather data of the agricultural meteorological experimental stations include the average temperature (Tavg), sunshine duration (SSD), and accumulated precipitation (PRE). The maize growth data include phenological stages, dry matter accumulation indices, yield components, varieties, and agronomic management measures. Among them, the phenological stages include sowing stage (SW), emergence stage (VE), jointing stage (V8), silking stage (R1), milking stage (R3), and maturity stage (R6); dry matter accumulation indices include grain dry matter (GY), vegetative organ dry matter (VODW), and total dry matter of the plant (TDW); and yield traits include ear number (EN), grain number (GN), and grain weight (GW). The crop management practices of the agricultural meteorological experimental stations are similar to local agricultural practices. That is, the varieties used (changed approximately every 2–3 years), sowing and harvesting dates, crop density, fertilization, and irrigation all conform to the actual situation of local farmers. The basic geographical information, phenology, irrigation, growth season climate conditions, and corn dry matter accumulation indices (yield traits) of each agricultural meteorological experimental station are shown in Tables S1–S3.

Selected average temperature (Tavg), accumulated effective precipitation (EP), and accumulated solar radiation (SRD) were the key climate factors. SRD was calculated based on sunshine duration (SSD) and the Ångström–Prescott equation [31], while accumulated effective precipitation (EP) was calculated using accumulated precipitation (PRE) [32]. To elucidate the effects of climate factors on maize during different growth stages, the entire maize growing season (GSw, VE-R6) was divided into four growth stages: seedling stage (GS1, VE-V8), ear development stage (GS2, V8-R1), grain formation stage (GS3, R1–R3), and grain filling stage (GS4, R3–R6).

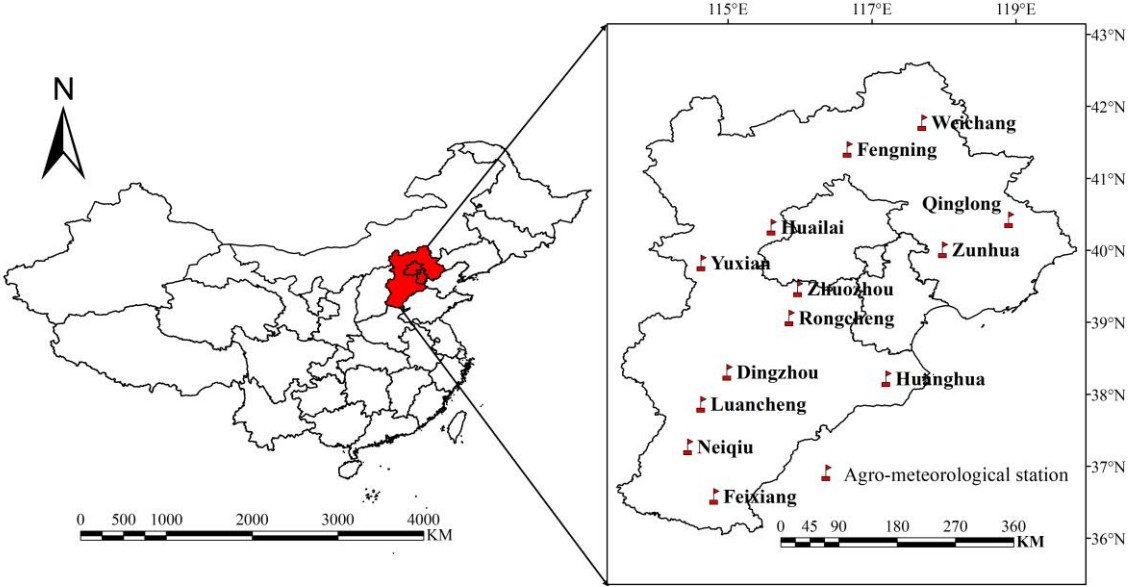

**Figure 1.** Study area and location of agrometeorological stations used in the study.

*2.2. Method*

The time series of regional average crop and meteorological data were obtained through the observed values at each station. Univariate linear regression was used to calculate the temporal trends of the dry matter accumulation indices and yield components of corn for the period 1981–2017, both regionally and at each station. The temporal trends (also known as climate tendency rates) of the main climate factors, including Tavg, EP, and SRD, during the entire growth period and each growth stage were also calculated.

In order to investigate the relationship between climate factors and maize dry matter accumulation indices (yield traits) during the entire growth stage and each growth stage in the study area, the time series of regional average crop data were detrended first to obtain the detrended series of crop data mainly influenced by climate change [33]. Secondly, linear detrending analysis was carried out on climate variables [34]. Finally, partial correlation analysis was used to investigate the correlation between detrended crop data and detrended climate factor data, statistically eliminating the covariate effects of climate factors [35]. Among them, the relationship between climate change in GS1 and GS2 stages and panicle number, the relationship between climate change in GS2 and GS3 stages and grain number, and the relationship between climate change in GS3 and GS4 stages and grain weight were analyzed separately.

Then, a multiple regression model was used to evaluate the sensitivity of maize dry matter accumulation indices (yield traits) at different growth stages to various climatic factors (Tavg, EP, and SRD) at each site. The model is formulated as follows:

$$Yd_t = \beta_0 + \beta_1 Year + \beta_{i2} Tavg_{ti} + \beta_{i3} EP_{ti} + \beta_{i4} SRD_{ti} + int \tag{1}$$

where $Yd_t$ represents the observed values (without detrending) of dry matter accumulation indices (yield traits) at each site in year $t$, $\beta_0$ is the intercept of the regression equation, $int$ is the regression error term, and $\beta_1$ represents the linear time trend at each site, which reflects the changes in crop data at the site during the study period due to variety, technology, and management improvements. $Tavg_{ti}$, $EP_{ti}$, and $SRD_{ti}$ are the observed values of a certain site's $i$-th growth stage (i.e., GS1, GS2, GS3, GS4, and GSw) in year $t$. $\beta_{i2}$, $\beta_{i3}$, and $\beta_{i4}$ are model parameters that represent the sensitivity of dry matter accumulation indices (yield traits) to changes in Tavg, EP, and SRD during the $i$-th growth stage. Furthermore, it is expressed in the percentage change in dry matter accumulation indices (yield traits) per year in the independent variable:

$$Q_{ij} = \left( \beta_{ij} / Yd_{mean} \right) \times 100\% \tag{2}$$

In the equation, $Yd_{mean}$ represents the average value of observed dry matter accumulation indices (yield traits) at a station from 1981 to 2017. Since this study only discusses the impact of various climate factors on maize and does not consider their interaction, it is assumed that the climate factors during each growth stage are independent. The impact of Tavg, EP, and SRD changes during the $i$-th growth stage at each site on the dry matter accumulation indices (yield traits) is represented separately using Equations (3)–(5):

$$\Delta T_{Yd\_} Tavg_i = Q_{i2} \times \Delta T_{Tavgi} \tag{3}$$

$$\Delta T_{Yd\_} T_{EPi} = Q_{i3} \times \Delta T_{EPi} \tag{4}$$

$$\Delta T_{Yd\_} T_{SDRi} = Q_{i4} \times \Delta T_{SDRi} \tag{5}$$

In the equation, $\Delta T_{Yd\_} Tavg_i$, $\Delta T_{Yd\_} T_{EPi}$, and $\Delta T_{Yd\_} T_{SDRi}$ represent the changes in maize dry matter accumulation indices (yield traits) caused by Tavg, EP, and SRD variations during the $i$-th growth stage from 1981 to 2017 (with the average value as reference). $\Delta T_{Tavgi}$, $\Delta T_{EPi}$, and $\Delta T_{SDRi}$ represent the changes in Tavg, EP, and SRD during the $i$-th growth

stage from 1981 to 2017. These changes were estimated by performing univariate linear regressions on Tavg, P, and SRD and multiplying the regression slopes by the duration.

According to Equation (1), the joint impact ($\Delta Yd\_clim_i$) of climate change during the *i*-th growth stage from 1981 to 2017 on the dry matter accumulation indices (yield trait) of a certain station is estimated as follows:

$$\Delta Yd\_clim_i = Q_{i2}\Delta T_{Tavgi} + Q_{i2}\Delta T_{EPi} + Q_{i2}\Delta T_{SDRi} \tag{6}$$

### 2.3. Statistical Analysis

Data analysis was conducted using Microsoft Excel 2019 (Redmond, WA, USA) and SPSS 21.0 (IBM, Inc., Armonk, NY, USA). Two-tailed *t*-tests were used to test for the statistical significance of partial correlations. In regression analysis, the significance of the regression equation was tested by F-test, and the significance of regression of each variable was tested by two-tailed *t*-test. To reduce the uncertainty caused by limited sample size, the bootstrap method was used to resample the data with 1000 iterations and obtain the median of regression coefficients. The spatial distribution maps were generated using inverse distance weighting (IDW) interpolation in ArcGIS 10.8 software, with a grid cell size of 90 m × 90 m.

## 3. Results

### 3.1. Changes in Climate Factors during the Entire Growth Period of Maize and within Each Growth Stage

The climatic trends of the three climate factors exhibited different spatial distribution characteristics in the entire corn growing season (GSw) (Figure 2). From 1981 to 2017, Tavg showed an upward trend in most areas, with only a downward trend in the northern region. EP showed an upward trend in the southern and northern regions, and a downward trend in the central and eastern regions. Except for the northwest region, where SRD showed an upward trend, it showed a downward trend in the rest of the regions. Therefore, the Tavg and EP of the corn growing season in the study area showed an overall upward trend, with climate trend rates of 0.11 °C/10a and 1.11 mm/10a, respectively (Table 1). The SRD showed an overall downward trend, with a climate trend rate of −8.09 MJ·m²/10a.

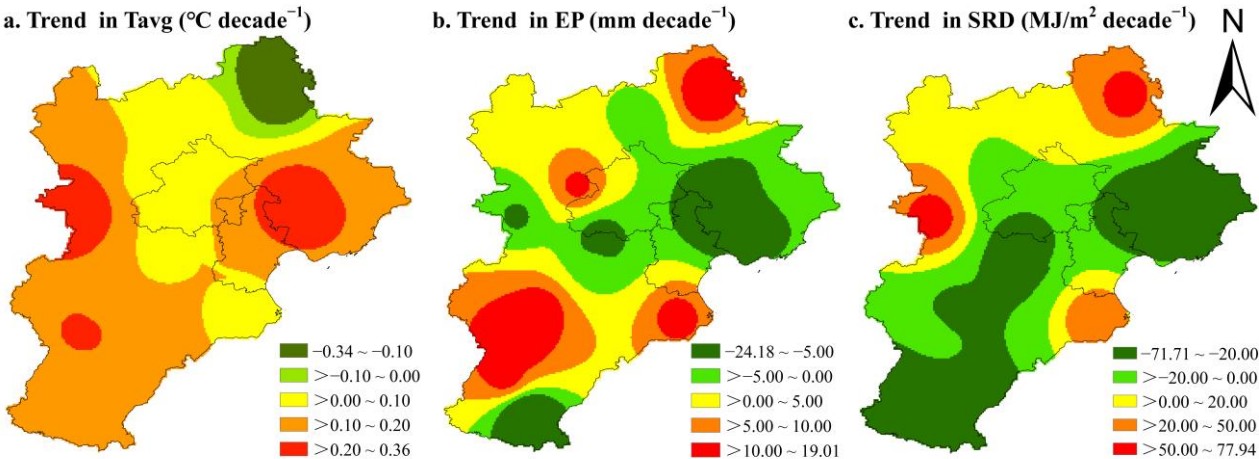

**Figure 2.** Spatial distribution of climate trend rates for average temperature (Tavg) (**a**), accumulated effective precipitation (EP) (**b**), and accumulated solar radiation (SRD) (**c**) during the entire growth period of maize from 1981 to 2017.

The climate conditions and trends during the full growth period and each growth stage of maize from 1981 to 2017 are shown in Table 1. On average, the Tavg during the GS2 stage was the highest, with an average of 25.22 °C, while the Tavg during the GS4 stage was the lowest. Significant warming was observed during the GS1 and GS2 stages, with

climate trend rates of 0.34 °C/10a and 0.36 °C/10a, respectively. There was no significant trend observed in EP during each growth stage, but the average EP during the maize nutritional growth stage (GS1–GS2) was 229.72 mm, accounting for 62.79% of the EP during the full growth period. The SRD during the nutritional growth stage was 1180.37 MJ·m$^{-2}$, which was higher than the 257.47 MJ·m$^{-2}$ during the reproductive growth stage (GS3–GS4). However, the SRD during the GS1 and GS2 stages showed a significant decreasing trend, while that during the GS3 and GS4 stages showed an increasing trend.

**Table 1.** Mean values and climate tendencies of average temperature (Tavg), accumulated effective precipitation (EP), and accumulated solar radiation (SRD) during the entire growth period and different growth stages of maize.

| Growth Stage | Tavg | | EP | | SRD | |
|---|---|---|---|---|---|---|
| | Average (°C) | Trend (°C/Decade) | Average (mm) | Trend (mm/Decade) | Average (MJ·m$^{-2}$) | Trend (MJ·m$^{-2}$/Decade) |
| GS1 | 23.56 ± 3.53 | 0.34 ** | 121.08 ± 24.96 | 1.83 | 751.10 ± 233.67 | −22.15 ** |
| GS2 | 25.22 ± 1.72 | 0.36 ** | 108.65 ± 24.07 | 3.35 | 429.78 ± 70.00 | −16.95 ** |
| GS3 | 23.87 ± 1.76 | 0.17 | 90.87 ± 39.56 | −6.79 | 480.83 ± 99.46 | 6.08 |
| GS4 | 20.33 ± 2.25 | −0.06 | 45.23 ± 13.42 | 3.09 | 442.07 ± 97.40 | 23.56 ** |
| GSw | 23.31 ± 2.22 | 0.11 | 365.83 ± 63.75 | 1.11 | 2103.27 ± 419.08 | −8.09 |
| GS1–GS2 | 24.14 ± 2.91 | 0.34 ** | 229.72 ± 34.27 | 5.18 | 1180.37 ± 287.06 | −39.10 ** |
| GS3–GS4 | 22.17 ± 1.86 | 0.01 | 136.11 ± 45.98 | −3.70 | 922.90 ± 141.97 | 29.64 ** |

Note: trends marked with "**" are significant at the 0.01 level.

### 3.2. Correlation between Maize Dry Matter Accumulation Indices (Yield Traits) and Climatic Factors

Under the combined effects of climate change and human management measures, both dry matter accumulation indices and yield traits showed a significant increasing trend from 1981 to 2017 ($p < 0.05$) (Figure 3). Among them, under the sole effect of climate change, climate change had a promoting effect on GW in each year, but the effect on EN was not significant.

During the entire growth period of maize (Figure 4), Tavg was negatively correlated with maize TDW, GY, and VODW, with Tavg and VODW reaching a significant level ($p < 0.05$). EP was positively correlated with maize TDW and GY, but negatively correlated with VODW, although the correlation was not significant. SRD was significantly positively correlated with maize TDW, GY, and VODW. Looking at each growth stage (Figure 4), Tavg was negatively correlated with TDW and VODW, and positively correlated with GY in GS4. In GS1, the negative correlation of Tavg with TDW, GY, and VODW reached a highly significant level ($p < 0.01$), and the correlation between Tavg and TDW, GY, and VODW decreased as the growth process progressed. In GS4, EP was significantly correlated with TDW and GY, but there was no significant correlation between EP and TDW, GY, and VODW in other growth stages. SRD was positively correlated with TDW, GY, and VODW in GS2–GS4, and the correlation between TDW, GY, and GS2–GS4 SRD was highly significant ($p < 0.01$).

According to Figure 5, there is no significant relationship between the three climatic factors and EN during the GS1–GS2 stages. EP at each growth stage is not significantly correlated with maize yield traits. SRD during the GS2–GS3 stage is significantly positively correlated with GN. Tavg and SRD during the GS4 stage are significantly positively correlated with GW, and the correlation between SRD and GW during the GS4 stage is extremely significant ($p < 0.01$).

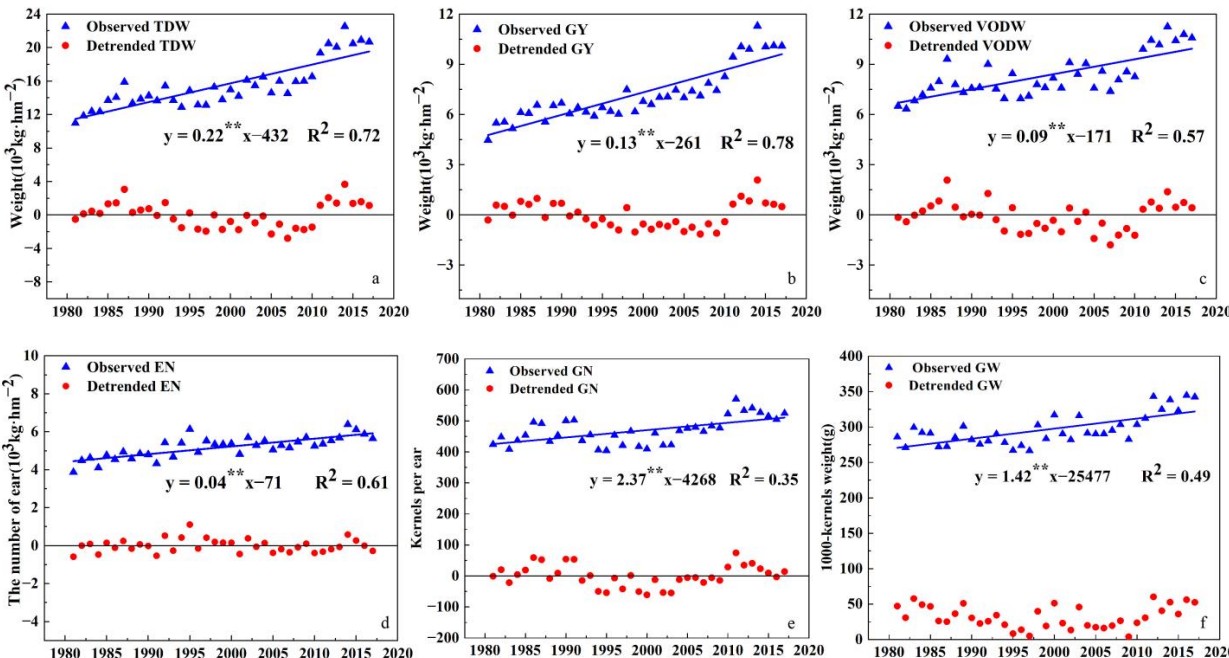

**Figure 3.** Observed and detrended values of dry matter accumulation indices and yield traits. '**' denotes significant trends at the 1% probability level.

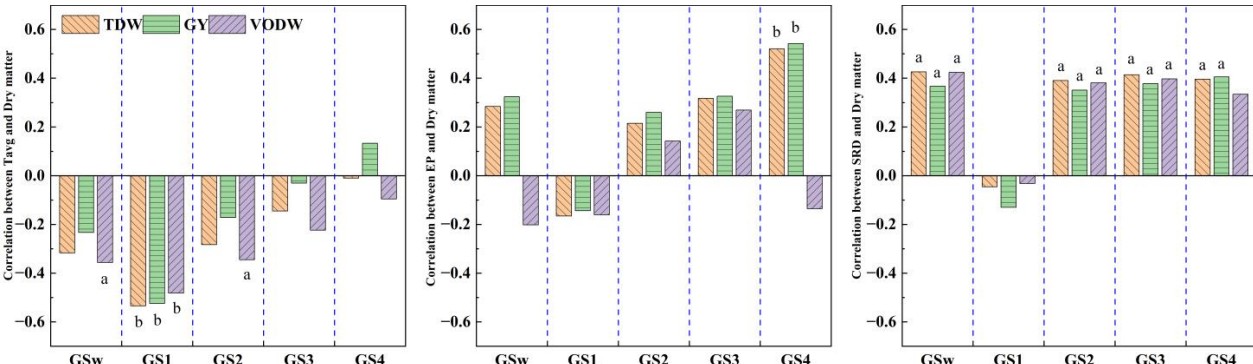

**Figure 4.** Partial correlation between climate factors and dry matter accumulation indices during the entire growth period and each growth stage of maize; correlations marked with "a" are extremely significant at the 0.05 level, and correlations marked with "b" are extremely significant at the 0.01 level.

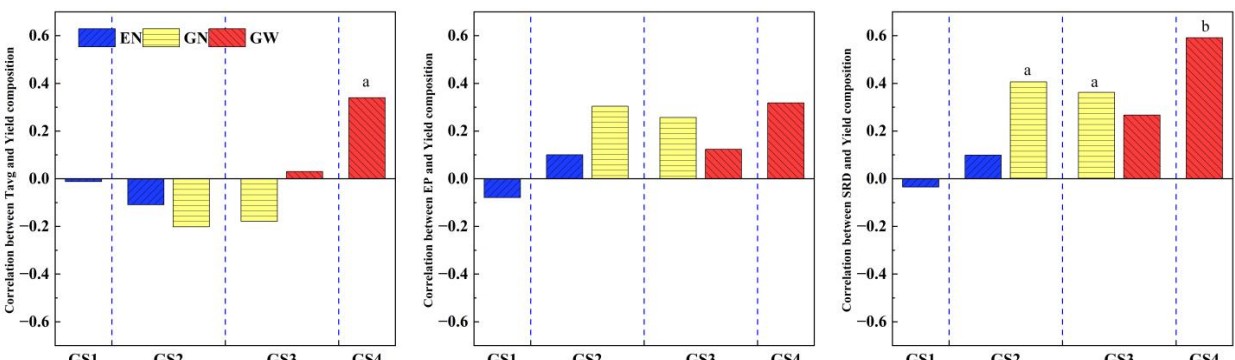

**Figure 5.** Partial correlation between climate factors and yield traits during corresponding growth stages. Correlations marked with "a" are extremely significant at the 0.05 level, and those marked with "b" are extremely significant at the 0.01 level.

### 3.3. Sensitivity of Maize to Climate Factor Changes

During the entire growth period of maize, an increase of 1 °C in Tavg led to an average decrease of 4.19%, 4.13%, and 4.25% in TDW, GY, and VODW, respectively (Figure 6A). Among them, TDW, GY, and VODW in the western and northeastern regions decreased by more than 5.0% (Figure 6a–c). With every increase of 10 mm in EP, TDW and GY increased by 0.10% and 0.23%, respectively, while VODW decreased by 0.01% (Figure 6B). TDW, GY, and VODW in the northwest and southeast regions increased by as much as 0.50%, while in the region from northeast to southwest, TDW, GY, and VODW decreased by 0.91%, 0.50%, and 1.25%, respectively (Figure 6d–f). An increase of 10 MJ·m$^{-2}$ in SRD led to an increase of 0.12%, 0.11%, and 0.12% in TDW, GY, and VODW, respectively (Figure 6C). Among them, TDW, GY, and VODW in the southern, central–western, and northeastern regions decreased by 0.45%, 0.44%, and 0.71%, respectively. After the dimensionless processing of the sensitivity of maize climatic factors for each station in the Beijing–Tianjin–Hebei region, it was found that TDW, GY, and VODW were more sensitive to Tavg than EP and SRD, with the number of stations being nine (69.23%), eleven (84.62%), and eight (61.54%), respectively. This indicates that maize is most sensitive to Tavg.

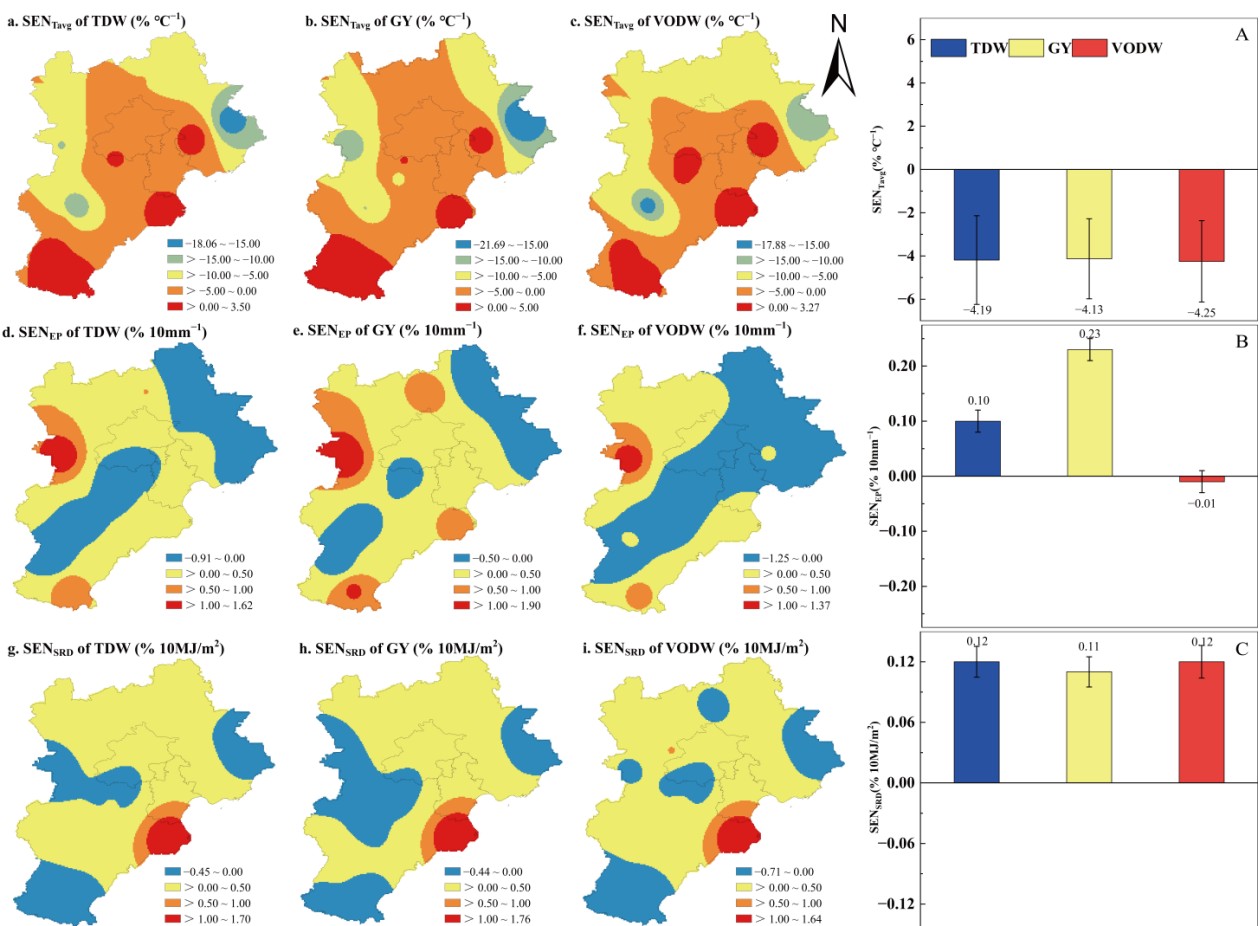

**Figure 6.** Spatial distribution of the sensitivity of dry matter accumulation indices to average temperature (SEN$_{Tavg}$) (**a–c**), accumulated effective precipitation (SEN$_{EP}$) (**d–f**), and accumulated solar radiation (SEN$_{SRD}$) (**g–i**) throughout the entire maize growth period. The bar graphs show the average sensitivity and standard error (represented by error bars) of dry matter accumulation indices to average temperature (**A**), accumulated effective precipitation (**B**), and accumulated solar radiation (**C**).

TDW, GY, and VODW are most sensitive to Tavg during the GS1 stage, with each 1 °C increase in Tavg resulting in a decrease of 3.28%, 3.86%, and 2.78% in TDW, GY, and VODW, respectively. During the GS2–GS4 stages, VODW is more sensitive to Tavg than GY

and TDW. TDW and GY are most sensitive to EP during the GS4 stage, with each 10 mm increase in EP resulting in an increase of 0.32% and 0.81% in TDW and GY, respectively. Conversely, VODW is significantly less sensitive to EP during the GS2–GS4 stages than TDW and GY. TDW, GY, and VODW are most sensitive to SRD during the GS4 stage, with each 10 MJ·m$^{-2}$ increase in SRD resulting in an increase of 0.43%, 0.48%, and 0.40%, respectively (Figure 7). Different yield traits have different sensitivities to climatic factors at different growth stages (Figure 8). GN is more sensitive to climatic factors during the GS3 stage than during the GS2 stage, while GW is more sensitive to Tavg during the GS3 stage and SRD during the GS4 stage, respectively.

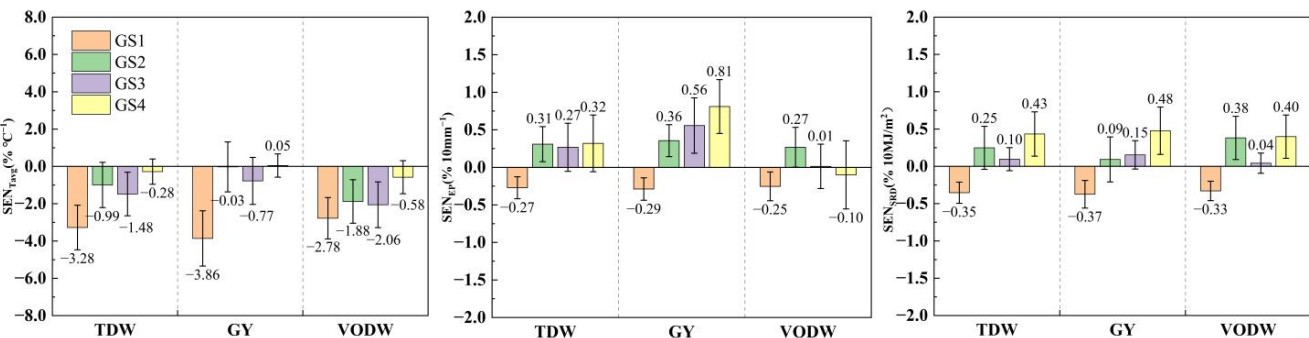

**Figure 7.** Sensitivity and standard errors (shown as error bars) of dry matter accumulation indices to climate factors at different growth stages.

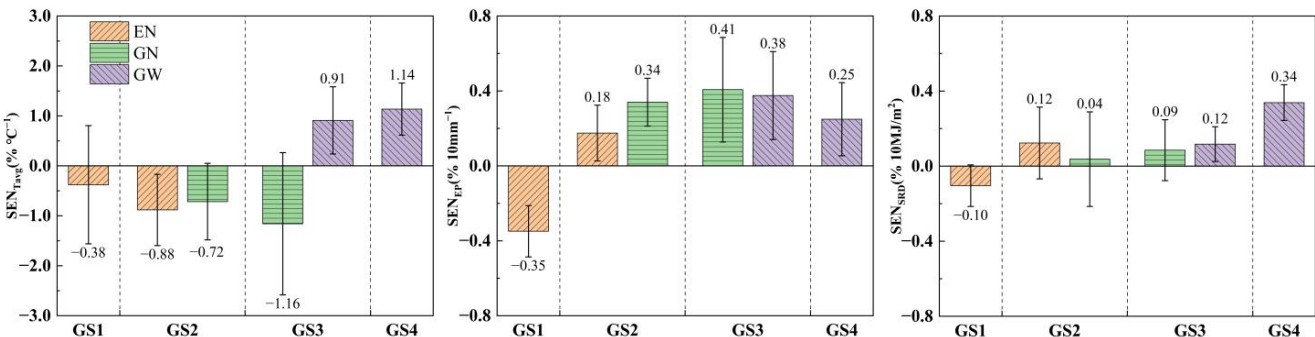

**Figure 8.** Sensitivity and standard error (represented by error bars) of yield traits to corresponding climate factors during different growth stages.

*3.4. The Impact of Climate Change from 1981 to 2017 on Maize Biomass Accumulation and Yield Traits*

The spatial distribution of changes in TDW, GY, and VODW induced by variations in different climate factors during the entire growth period of maize was generally consistent (Figure 9). Under the joint action of the three climate factors, the regions in the central–western and central–eastern parts of China showed a decreasing trend, while the regions from north to south showed an increasing trend (Figure 9A–C). Among them, TDW and GY decreased by an average of 0.18% and 0.94%, respectively, while VODW increased by an average of 0.53%. The trend of Tavg changes resulted in a decrease in TDW, GY, and VODW in most regions, but the reduction was within 5% (Figure 9a,d,j). Except for the central–eastern region, the trend of EP changes caused a decrease in TDW, GY, and VODW within 5% (Figure 9b,c,h). The trend of SRD changes increased TDW, GY, and VODW in most areas, with an increase of 5% to 30% (Figure 9c,f,i).

The effects of climate change on maize TDW, GY, and VODW varied across different growth stages (Figure 10). The negative impact of climate change (especially the increase in Tavg) on TDW, GY, and VODW was most evident in the GS1 stage, with reductions of 5.05%, 8.19%, and 2.35%, respectively. Climate change in the GS3–GS4 stages (especially the significant increase in SRD) increased the final weight of TDW, GY, and VODW (Figure 10).

The upward trend of SRD in the GS3 stage resulted in increases of 3.52%, 4.71%, and 2.50% for TDW, GY, and VODW, respectively, while the upward trend of SRD in the GS4 stage led to increases of 5.77%, 5.81%, and 5.81% for TDW, GY, and VODW, respectively. In addition, the upward trend of SRD in the GS3 stage increased the number of grains by 5.00%, and the upward trend of seed weight in the GS4 stage increased by 2.84%. The effects of other climate factors on yield traits during the GS3–GS4 stages were relatively small (Figure 11).

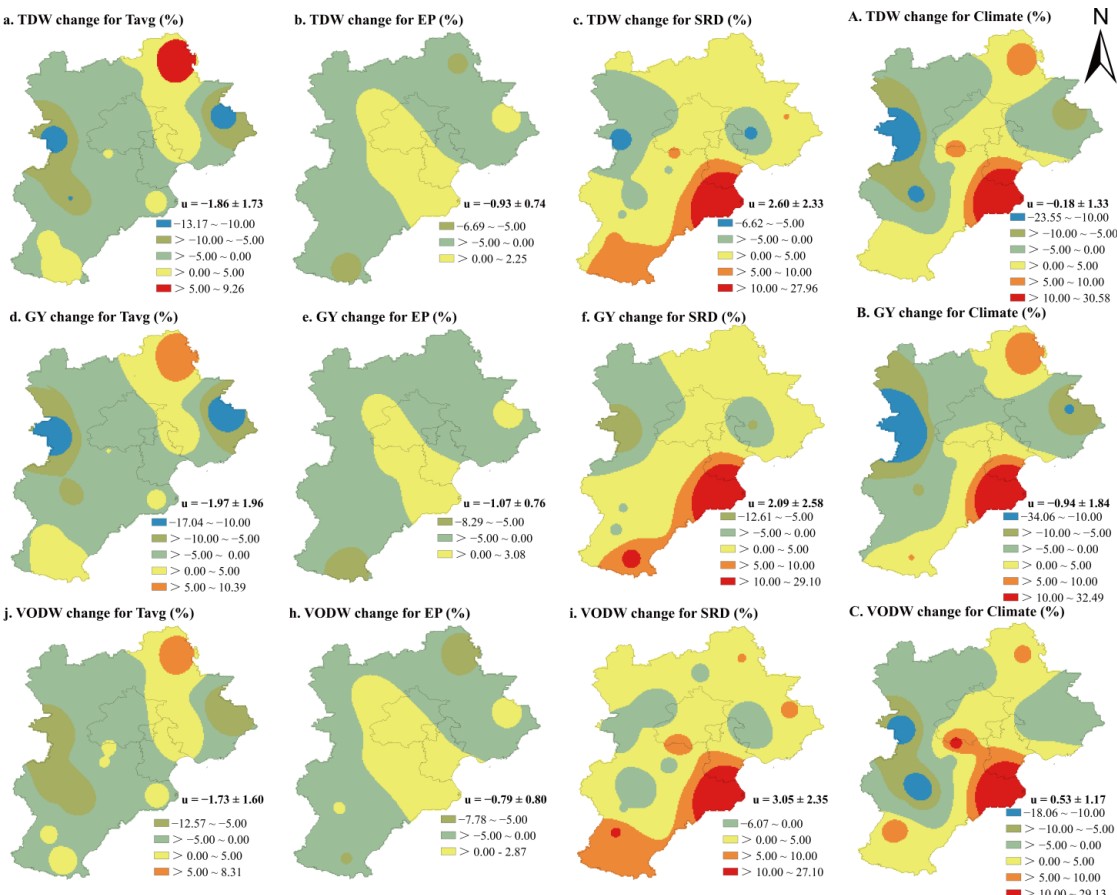

**Figure 9.** The spatial distribution of changes in plant total dry weight (TDW) (**a–c**), grain yield (GY) (**d–f**), and vegetative organ dry weight (VODW) (**j–i**) under the influence of mean temperature (Tavg), accumulated effective precipitation (EP), and accumulated solar radiation (SRD) throughout the whole maize growth period. Panels A-C show the spatial distribution of changes in TDW (**A**), GY (**B**), and VODW (**C**) under the combined effects of all three climate factors.

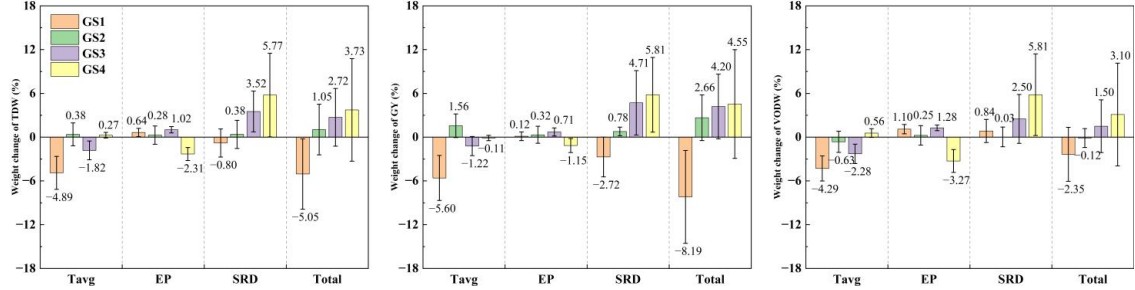

**Figure 10.** Effects of climate variables including average temperature (Tavg), accumulated effective precipitation (EP), accumulated solar radiation (SRD), and their combined impact on maize dry matter accumulation indices at different growth stages. Error bars represent estimated standard errors.

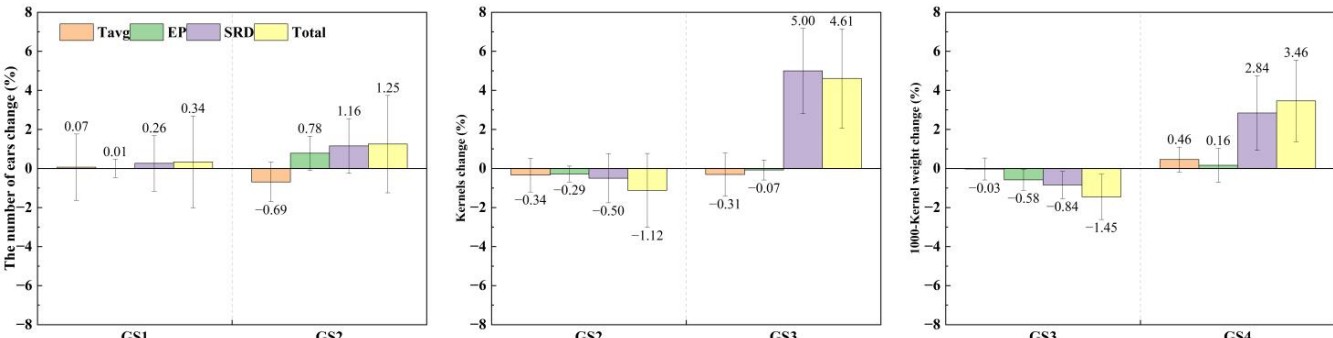

**Figure 11.** Effects of average temperature (Tavg), accumulated effective precipitation (EP), accumulated solar radiation (SRD), and their combined effects on maize yield traits during corresponding growth stages. Error bars represent estimated standard errors.

## 4. Discussion

### 4.1. Climate Warming inhibited the Growth and Development of Maize during the Vegetative Growth Stage

Climate warming is a key feature of climate change [1]. Temperature affects crop biomass accumulation and yield through physiological processes such as photosynthesis and respiration [36]. Numerous studies have shown that climate warming during the maize growth period has a negative impact on maize yield [37–39]. The results of this study indicate that higher temperatures not only have a negative impact on maize grain yield, but also on above-ground dry weight and nutrient organ dry weight. However, the negative impact on maize grain yield is more pronounced. Analyzing by growth stage, the significant inhibition of maize growth and development during the seedling stage due to temperature increase is the main reason for this (Figure 10). This is because increased temperatures during the seedling stage accelerate maize growth [30], but the solar radiation decreases significantly during this period (Table 1), which is unfavorable for maize photosynthesis, leading to a negative effect on maize production. In addition, previous studies have shown that climate warming increases the water demand of maize [40]. Maize is more sensitive to temperature when water is scarce than when it is sufficient [41]. Moreover, the maize growing season in the Beijing–Tianjin–Hebei region is lacking in precipitation resources [42], and the significant increase in temperature during the seedling stage makes maize more sensitive to temperature than during other growth stages. This study also found that maize becomes less sensitive to temperature and more sensitive to water and radiation during the silk stage (Figure 7). This is because the silk stage is a critical period for the formation of maize kernels and requires stable photosynthesis to ensure the supply of assimilates, making it more sensitive to water and light [43]. Furthermore, the analysis of temperature sensitivity for the number and weight of kernels during the silk stage showed that continuous warming during this period would increase kernel weight while reducing the number of effective kernels, and may not necessarily increase total yield. Therefore, we believe that the negative impact of climate warming on maize is mainly due to the inhibition of the growth and development of maize nutrient organs before the silk stage.

### 4.2. The Increase in Cumulative Solar Radiation after Tasseling Promotes the Growth and Development of Maize

According to some studies, atmospheric pollution has significantly reduced solar radiation during the corn growing season in recent years, causing serious effects on corn growth and development, with a greater impact than temperature increase [15]. However, in this study, solar radiation during the corn growing season did not show a significant downward trend, and changes in solar radiation had a positive effect on corn growth and development. This result is also confirmed in the study by Stanhill and Cohen [44], indicating that a small reduction in solar radiation can increase crop yield. From the perspective of different growth stages, the significant increase in solar radiation during the



later stages of corn growth promotes corn growth and development, especially in terms of grain number and weight, which compensates for the adverse effects of the decreased solar radiation during the earlier stages of growth. Liu et al. [45] also obtained consistent results with our study, showing that an increase in accumulated solar radiation during the corn grain-filling period improved the negative effect of solar dimming and promoted an increase in corn yield. In addition, this study found that corn is most sensitive to temperature, but the effect of temperature on corn is not greater than that of solar radiation. This is because the significant increase in solar radiation during the later stages of growth was coupled with corn's sensitivity to solar radiation in time and space, making the effect of solar radiation fully demonstrated [46]. Some studies have shown that insufficient light and heat resources during the corn growing season are still among the main limiting factors for increasing corn yield in the North China Plain [47]. From this study, it can be seen that against the background of climate change, solar radiation during the later stages of corn growth will continue to increase, which will help increase corn yield in the Beijing–Tianjin–Hebei region. However, the increase in solar radiation may be caused by the use of long-grain-filling-period varieties, but this study did not accurately calculate the increase in solar radiation caused by the extension of the growth period. Therefore, this should be considered in future studies to quantify the contribution of climate change to corn growth and development.

*4.3. The Impact of Climate Change on the Growth and Development of Maize Exhibits Spatial Variability*

Previous studies have indicated that the relationship between crop yield and climate varies across different spatial scales in China. Some climate–yield relationships that are unclear at larger scales exhibit significance at smaller scales [37,48]. This study used point data for spatial analysis and found significant spatial variability in the sensitivity of maize yield to the same climate factors (Figure 6). Butler and Huybers [49] also obtained similar results, revealing large regional differences in the temperature sensitivity of US maize yields. Our results indicate that, under the combined effects of multiple climate factors, climate change has a negative impact on maize in western and eastern regions, while it has a positive impact in other regions. The dominant climate factor contributing to positive effects is solar radiation, while in the western regions with negative effects average temperature is the dominant factor (Figure 9). Xu et al. [50] also confirmed that there are significant spatial differences in the distribution of the dominant influencing factors for maize yield potential across different agricultural regions in China, with 47% of counties being mainly affected by changes in solar radiation and 16% being mainly affected by changes in temperature. Studies have shown that the negative impacts of climate warming can be alleviated through effective irrigation [51,52]. Therefore, the western region of Beijing–Tianjin–Hebei could consider strengthening irrigation during the seedling stage, or it could also improve soil water storage capacity and effective water content through measures such as straw returning and deep loosening, in order to reduce the adverse effects of climate warming.

**5. Conclusions**

Our study indicates that the negative effects of climate change are mainly distributed in the central-western and central-eastern regions of Beijing–Tianjin–Hebei. Among the three investigated climate factors, maize is most sensitive to average temperature, but changes in solar radiation have a greater impact on maize than temperature and precipitation. The significant increase in solar radiation after the maize tasseling stage effectively promoted the increase in maize grain number and weight, alleviating the adverse effects of temperature increase in the early growth stage of maize in the Beijing–Tianjin–Hebei region. However, in the long term, the trend of climate warming will continue to intensify, and it is necessary to consider strengthening irrigation in the seedling stage or increasing the soil's effective water content to mitigate the adverse effects of climate warming in the western region

of Beijing–Tianjin–Hebei. The research results can provide a basis for decision-making regarding maize production against the background of future climate change.

**Supplementary Materials:** The following supporting information can be downloaded at: https://www.mdpi.com/article/10.3390/agronomy13051284/s1, Table S1. Geographical information of agricultural meteorological experimental stations in the Beijing–Tianjin–Hebei region and climatic conditions during the maize growing season; Table S2. Dataset of phenological stages for maize across agricultural meteorological stations in the Beijing–Tianjin–Hebei region; Table S3. Management practices and maize growth and development at each agricultural meteorological experimental station in the Beijing–Tianjin–Hebei region.

**Author Contributions:** Conceptualization, Z.W. and Y.C.; methodology, Z.W., D.B., X.D., Z.G. and C.L.; software, D.B., X.D., Z.G., G.L., Q.Y. and A.J.; validation, D.B., Z.G. and G.L.; formal analysis, Z.W., X.D., Z.G., C.L. and G.L.; investigation, Z.W., C.L. and Y.C.; resources, Z.W. and C.L.; data curation, Z.W., D.B., Q.Y. and A.J.; writing—original draft preparation, Z.W.; writing—review and editing, Z.W., D.B., X.D., Z.G., C.L. and Y.C.; visualization, Z.W. and D.B.; supervision, Y.C.; project administration, X.D.; funding acquisition, Y.C. All authors have read and agreed to the published version of the manuscript.

**Funding:** This research was funded by the National Key Research and Development Program of China (No. 2017YFD0300903, 2017YFD0300908); The Corn Industry Technology System of Hebei Province (HBCT2018020101, HBCT2018020202); and The Key Research and Development Program of Hebei Province (20326414D).

**Data Availability Statement:** Data recorded in the current study are available in all tables and figures of the manuscript.

**Conflicts of Interest:** The authors declare no conflict of interest.

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
