# Peer review of "An Increase in Solar Radiation in the Late Growth Period of Maize Alleviates the Adverse Effects of Climate Warming on the Growth and Development of Maize"

_agronomy, doi:10.3390/agronomy13051284_

Round 1

Reviewer 1 Report

Comments in the file.

Author Response

Dear reviewer,

Thank you very much for giving us the opportunity to revise our manuscript. We appreciate your positively constructive comments and suggestions on our manuscript entitled “The Increase of Solar Radiation in the Late Growth Period of Maize Alleviates the Adverse Effects of Climate Warming on the Growth and Development of Maize (agronomy-2314957)”. Those comments are valuable and very helpful for revising and improving our paper, and we have made correction which we hope meet with approval. Revised portion are marked in the “Revised Manuscript with Track Changes” file. Please let us know if more revisions are needed. Thanks again. We look forward to hearing from you.

Sincerely yours,

Zhongbo Wei on behalf of all the authors

The main corrections are as follows: 

Point 1: In general, the paper is very interesting and results looks consistent. However, the methodology is described in a very confusing way. Therefore, I recommend the authors re-write all the methodology in a simple and clearer way.

Response 1: We have re-write the methodology in Line 166-271 in the “Revised Manuscript with Track Changes” file.

Reviewer 2 Report

The manuscript explores a topic that is of interest to readers in the field of Agronomy. However, I have identified several weaknesses that need to be addressed prior to publication. Firstly, the abstract needs to be improved to better align with the title of the work. Additionally, abbreviations should be defined the first time they are used to enhance clarity and readability.

 In the materials and methods section, there are some missing information that need to be added. Specifically, it is not clear which varieties of corn were used in each site, and the planting dates and different phenological stages are not specified. It would also be helpful to know whether irrigation was used in the sites throughout the years, and this information should be included as supplementary material. Furthermore, it would be beneficial to include meteorological information collected at each site during the study period.

 Another important consideration is the statistical analyses used in the study. It is not clear which statistical tests were carried out or what the design of the analysis was. 

 Overall, addressing these issues will significantly improve the manuscript and increase its potential for publication in a reputable journal.

Author Response

请参阅附件。

Reviewer 3 Report

Dear Authors, 

I have read your submitted manuscript and although I appreciate your hard work in investigations and experimental fields. 

After minor revisions (especially with placing the graphs) I recommend the manuscript for acceptance. 

With the aim of better understanding and easier reading of a scientific paper, I recommend placing graphs and text as close as possible.

Author Response

Dear reviewer,

Thank you very much for your recognition of our work. We appreciate your positively constructive comments and suggestions on our manuscript entitled “The Increase of Solar Radiation in the Late Growth Period of Maize Alleviates the Adverse Effects of Climate Warming on the Growth and Development of Maize (agronomy-2314957)”. Those comments are valuable and very helpful for revising and improving our paper, and we have made correction which we hope meet with approval. Revised portion are marked in the “Revised Manuscript with Track Changes” file.  Please let us know if more revisions are needed. Thanks again. We look forward to hearing from you.

Sincerely yours,

Zhongbo Wei on behalf of all the authors

The main corrections are as follows:

Point 1: With the aim of better understanding and easier reading of a scientific paper, I recommend placing graphs and text as close as possible.

Response 1: Thank you for suggestion. We have re-arranged the graphs in the manuscript.

Point 2: “field experiment data from 1981-2017 in the Beijing-Tianjin-Hebei region and relevant meteorological data from 13 national agricultural meteorological stations were analyzed to investigate the relationship between climate change and maize growth and yield.” This part of the sentence is more about Materials and Methods. Please rewrite this sentence.

Response 2: We have revised this sentence in line 96-98 in the “Revised Manuscript with Track Changes” file.

Round 2

Reviewer 1 Report

Finally, partial correlation analysis method was used to analyze the correlation between detrended crop data series and detrended climate factor series, which statistically eliminated the influence of climate factors

The correlations detrend between detrended crop data series and detrended climate factor series does not eliminated the influence of climate factors – it eliminates the temporal trend of the variables.

By the sentence above, I am not sure if the authors really use dry matter accumulation    without detrending, see the comments bellow.

I does not understand the use of ??? without detrending. For my understanding, here the authors should use the detrend variables (for climate and growth variables),  to establish the relationship between the dry matter accumulation and climate variables – i.e., eliminating the temporal trends in yield and climate variables for the establishing of the statistical model (eq. 1). This is necessary because trends in yield can be explained by trends in climate variables and by changes in management and genetic materials used, and you want to establishing the relation between yield and climate only.

After establishing these relationships, between detrended yield and detrended  climate variables, the impacts of climate trends on yield would be evaluated by subsequent equations.

Reviewer 2 Report

The manuscript presented has undergone significant improvements compared to its previous version. 

In particular, the authors have carefully considered the reviewers' comments and responded to them effectively. They have addressed the limitations identified in the previous version and clarified some points that were not entirely clear. Furthermore, they have added new relevant information that has further strengthened their arguments.

In my opinion, the work presented now meets the necessary standards for publication. The changes made have improved the clarity and quality of the text and strengthened the validity and relevance of the results. I am convinced that readers will find this work informative and interesting, and it will significantly contribute to the advancement of knowledge in this field.

Author Response

Dear reviewer,

Thank you for your affirmation. We will continue our efforts.

Kind regards,

Zhongbo Wei on behalf of all the authors.

Round 3

Reviewer 1 Report

Even though I have a different view on how to process the data, the authors presented a valid, peer-reviewed, alternative way to obtain the statistical model.

Author Response

(The authors gave the same response as above.)
